# The Antithetic Roles of IQGAP2 and IQGAP3 in Cancers

**DOI:** 10.3390/cancers15041115

**Published:** 2023-02-09

**Authors:** Fei Song, Qingqing Dai, Marc-Oliver Grimm, Daniel Steinbach

**Affiliations:** 1Department of Urology, Jena University Hospital, 07740 Jena, Germany; 2Department of Internal Medicine IV (Gastroenterology, Hepatology, and Infectious Diseases), Jena University Hospital, 07740 Jena, Germany

**Keywords:** IQGAP, cancer, pathway, signaling, biomarker

## Abstract

**Simple Summary:**

The IQ motif-containing GTPase-activating protein family is comprised of three signal scaffolding proteins that regulate a variety of biological functions by aiding signal transduction in cells. IQGAPs induce numerous cancer-related processes, including proliferation, apoptosis, migration, invasion, and angiogenesis. In comparison to IQGAP1, IQGAP2 and IQGAP3 were less researched. In this review, we comprehensively reviewed the significant roles of IQGAP2 and IQGAP3 in cancer-associated pathways as well as the role in carcinogenesis and progression of different cancer entities.

**Abstract:**

The scaffold protein family of IQ motif-containing GTPase-activating proteins (IQGAP1, 2, and 3) share a high degree of homology and comprise six functional domains. IQGAPs bind and regulate the cytoskeleton, interact with MAP kinases and calmodulin, and have GTPase-related activity, as well as a RasGAP domain. Thus, IQGAPs regulate multiple cellular processes and pathways, affecting cell division, growth, cell–cell interactions, migration, and invasion. In the past decade, significant evidence on the function of IQGAPs in signal transduction during carcinogenesis has emerged. Compared with IQGAP1, IQGAP2 and IQGAP3 were less analyzed. In this review, we summarize the different signaling pathways affected by IQGAP2 and IQGAP3, and the antithetic roles of IQGAP2 and IQGAP3 in different types of cancer. IQGAP2 expression is reduced and plays a tumor suppressor role in most solid cancer types, while IQGAP3 is overexpressed and acts as an oncogene. In lymphoma, for example, IQGAPs have partially opposite functions. There is considerable evidence that IQGAPs regulate a multitude of pathways to modulate cancer processes and chemoresistance, but some questions, such as how they trigger this signaling, through which domains, and why they play opposite roles on the same pathways, are still unanswered.

## 1. Introduction

IQ motif-containing GTPase-activating proteins (IQGAPs) are a family of evolutionarily conserved proteins found in a broad range of protists, fungi, and animal cells. In humans, there are three IQGAP proteins, IQGAP1, IQGAP2, and IQGAP3, which share a similar domain structure and sequence homology. These scaffold proteins are often located between cell junctions of epithelial cells, and interact with components of the cytoskeleton, cell adhesion, and signaling molecules that regulate various cellular processes [1,2,3,4,5].

The multidomain structure of IQGAPs facilitates the formation of protein complexes necessary for cellular functions. Since its identification in human osteosarcoma tissue in 1994 [6], IQGAP1, which is ubiquitously expressed, has attracted considerable interest as the best-characterized isoform of IQGAPs. More than a hundred interacting proteins with diverse functions have been characterized [7], which have been subsequently implicated in a number of cellular processes, including cell proliferation [8,9], cytokinesis [10,11], vesicle trafficking [12,13], cell migration [14,15], and cytoskeletal dynamics [16,17].

As the first identified member of the IQGAP protein family, IQGAP1 has gained the greatest interest. Upregulated IQGAP1 was detected in multiple cancer entities, including breast [18,19], colorectal [20,21], esophageal [22,23], gastric [24,25], head and neck [26,27], ovary [28], pancreases [29], and thyroid cancer [30]. Patients with increased levels of IQGAP1 have a worse prognosis, demonstrating the prognostic value of IQGAP1 in many cancers. Hundreds of studies have clarified and summarized that IQGAP1 is able to interact with and activate different oncogenic pathways [7], including the MAPK, RAC1/CDC42, Wnt/β-catenin, PI3K, Hippo, and TGF-β pathway, as well as ultimately promoting tumor proliferation, migration, and invasion.

IQGAP2 was identified in 1996 as a large cytoplasmic scaffold protein that is expressed predominantly in the liver, but also in the prostate, kidney, stomach, testis, and platelets [31]. Despite its 62% sequence homology with other IQGAPs, the majority of evidence demonstrated that IQGAP2 acted as a tumor suppressor in malignancies [32,33,34,35,36,37,38,39,40]. In contrast, IQGAP3 is an oncogene [41,42,43,44,45]. It correlates with poor cancer prognosis [46,47,48,49]. IQGAP3 was identified in 2007, is more strongly expressed in the brain, and is important for the regulation of neurite growth in neurons [50]. IQGAP3 is necessary for proper cell proliferation and motility in addition to governing mitotic progression, genomic integrity, and stability [51,52].

Despite their apparent similarities, the expression patterns of IQGAPs are inconsistent (Table 1).

Compared with IQGAP1, we have little knowledge on IQGAP2 and IQGAP3. The last review published about these two IQGAPs was in 2015 [54], and our understanding of IQGAP2 and IQGAP3 has further deepened in the past few years. In this review, we summarize the different signaling pathways affected by IQGAP2 and IQGAP3, and the antithetic roles of IQGAP2 and IQGAP3 in different types of cancer.

## 2. The Molecular Domains of IQGAPs

IQGAPs share a high degree of homology in amino acid sequence and domain structure. The six major functional domains allow them to bind to several partners, thus altering the spatial and temporal distribution of distinct signal transduction complexes (Figure 1).

The N-terminal calponin-homology domain (CHD) of IQGAPs, which is also present in several actin-binding proteins, is able to directly interact with actin [31,50,55,56]. This induces rearrangements of the actin cytoskeleton, and regulates cell division, cell migration, and the stability of cell–cell interactions.

The coiled-coil repeat region (CC) domain, which consists of hydrophobic and charged amino acids in a repeated pattern, facilitates IQGAPs to bind to the ezrin-radixin-moesin (ERM) protein family [57,58]. This reinforces the role of IQGAPs as cytoskeletal regulators and signal transduction hubs.

The polyproline protein–protein domain contains two functionally conserved tryptophans. These are responsible for the interaction with classical MAP kinases (MAPK), and stimulate downstream signaling pathways, thus promoting tumor growth, progression, and invasion [59,60].

There are four tandem isoleucine-glutamine (IQ) motifs in the IQ domain, which interact in a calcium-dependent manner with calmodulin, which regulates several cellular processes [61,62].

The IQ domain is followed by a GTPase activation-related structural domain (GRD) that is extremely identical to the functional components of Ras GTPase-activating proteins (GAPs), and interacts with the Rho family members, small GTPases, such as Rac1 and Cdc42, altering their GTP activity [63,64]. This is, among other things, critical for the polymerization of actin filaments throughout the progression of cancer [65].

Lastly, IQGAPs possess an exclusive C-terminal RasGAP domain. By binding to a range of proteins, including E-cadherin and beta-catenin, this domain contributes considerably to mediate cell–cell adhesion, cell polarization, and directional migration [66].

**Figure 1 cancers-15-01115-f001:**
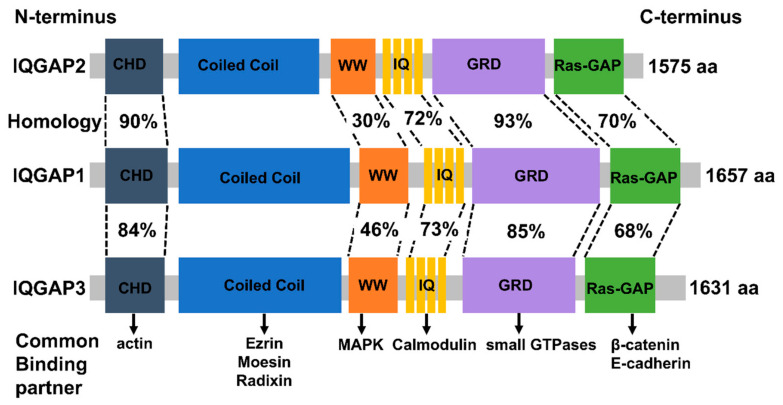
Schematic depiction of the domain organization and amino acid homology of IQGAPs and their common binding partners (Permission obtain from OA journal: [67]).

## 3. IQGAPs Mediate Multiple Key Pathways

Multiple signaling pathways, such as the MAPK/ERK, receptor tyrosine kinase (RTK)-activated phosphatidylinositol 3-kinase/AKT (PI3K-AKT), transforming growth factor β (TGF-β), and Wnt/-catenin pathways, are aberrantly activated in malignancies, leading to unregulated cell division, differentiation, proliferation, motility, and apoptosis (Figure 2). Increasing research indicates that altered IQGAP expression affects cancer progression.

### 3.1. MAPK Signaling

IQGAP2 and IQGAP3 are both correlated with the MAPK/ERK pathway, which is one of the most crucial signaling pathways in the progression of cancer, promoting tumor growth and metastasis [68,69,70]. Zhu Y. et al. established immortalized gastric epithelial cells transformed by cagA, an important virulence factor closely related to gastric cancer patients, and found it was achieved through abnormal activation of the ERK1/2 MAPK cascade. The aberrant activation of the MAPK cascade then increased the expression level of IQGAP2, as confirmed by real-time PCR and Western blot analysis [71]. However, the knockdown of IQGAP2 in breast cancer and bladder cancer elevated the phosphorylation of MEK1/2 and ERK1/2, which resulted in promoting EMT [40]. A further pull-down assay indicated that IQGAP2 levels in the cells can regulate IQGAP1-mediated ERK activation. Overall, this evidence proved IQGAP2 is related to MAPK signaling, but the mechanism needs to be further studied in detail.

The activation of ERK mediated by IQGAP3 was found in gastric cancer [47], breast cancer [72,73], lung cancer [44], and bladder cancer [43], which is more likely achieved through the interaction between IQGAP3 and RAS, while IQGAP2 had no effect on the Ras activity within cells [45]. Nojima et al. found that IQGAP3 specifically interacted with the active form of Ras. The Ras activity was significantly downregulated after knockdown of IQGAP3, following completely abrogated activation of ERK2 in mouse mammary gland epithelial cells. This suggests that Ras/ERK is a functional downstream effector of IQGAP3 [51]. Meanwhile, the evidence that Ras signaling pathway key components Erbb2, Erbb5, Fgfr2, Fgfr3, Met, and Ras were enriched in isthmus stem cells with high IQGAP3 expression confirmed this positive interaction between IQGAP3 and Ras [74]. However, Chen et al. discovered the reverse outcome in DLBCL cells, where shRNA-directed downregulation of IQGAP3 led to increased RAS activity [75].

### 3.2. Wnt Signaling

The Wnt signaling pathway is involved in numerous cellular functions, including proliferation, differentiation, migration, polarization, and self-renewal during development [76], but also results in abnormal cell proliferation and the promotion of tumors [77,78].

Schmidt et al. identified a multiprotein β-catenin-E-cadherin-IQGAP1-IQGAP2 scaffold in hepatocytes. Further analysis detected the activation of the WNT pathway, defined by the loss of E-cadherin, activation of β-catenin, and overexpression of the β-catenin nuclear target cyclin D1, in an IQGAP2-deficient mice model [35]. By using Affymetrix microarray technology, it was demonstrated that the Wnt/-catenin signaling pathway is the most commonly altered pathway in HCC tumors in the IQGAP2-deficient mouse model [79]. A similar effect was found in ovarian cancer, where IQGAP2 strongly inhibited the expression and nuclear translocation of β-catenin. Notably, IQGAP2 inhibited the Wnt3a-induced transcriptional activity of β-catenin in ovarian cancer, indicating that it is a suppressor of Wnt signaling [34].

### 3.3. PI3K-AKT Signaling

The Akt pathway, also known as the PI3K-Akt pathway, is involved in essential biological activities, such as protein synthesis, cell proliferation, and apoptosis [80,81].

It has been discovered that IQGAP2 increases E-cadherin expression and inhibits EMT via a reduction of Akt activation in prostate cancer [33]. Further analysis in gastric cancer revealed the underlying mechanism, whereby IQGAP2 interacted with SHIP2, and increased its phosphatase activity, deactivating Akt and decreasing EMT [38]. In a recent study, the increased expression of IQGAP2 significantly decreased the amount of VEGF-A in breast cancer cells, and the consequent phosphorylation of VEGFR2 in endothelial cells, which resulted in the phosphorylation of AKT molecules [82].

IQGAP3 was found to interact with protein kinase C delta (PKCδ), a binding factor of protein kinase C alpha (PKCα), to block the association between PKCδ and PKCα. This releases PKCδ, and activates PKCα through phosphorylation, resulting in the activation of PI3K/AKT signaling pathways to promote HCC cell proliferation [83]. In addition, as one of the three isoforms of class II PI3Ks, PIK3C2B, which can activate AKT signaling pathways, was upregulated at the protein and mRNA transcription levels following exogenous overexpression of IQGAP3 in colon cancer, resulting in promoting invasiveness [42].

### 3.4. TGF-β Signaling

TGF-β signaling regulates the proliferation, differentiation, and apoptosis of numerous cancers [84,85].

The upregulated TGF-β-responsive luciferase activity, phosphorylation of Smad2, Smad3, and downstream proteins of the TGF-β signaling pathway were detected in response to IQGAP3 overexpression in HCC cell lines. This resulted in induction of EMT in HCC. SB431542, a TGF inhibitor, inhibited the effects of IQGAP3 on HCC cell motility and invasion, suggesting that activation of TGF-β signaling is a crucial mediator for IQGAP3-induced HCC metastasis [86].

### 3.5. NF-κB Signaling

The NF-κB signaling plays an important role in inflammatory processes and is key in the development and dysfunction of the immune system [87,88]. The NF-κB family, one of the most important transcription factors linking chronic inflammation to cancer, consists of five members: RelA (p65), c-Rel, RelB, NF-κB 1 (p50), and NF-κB 2 (p52).

It has been reported that IQGAP2 physically interacts with p65 [89,90], which induces genes implicated in inflammation, cell proliferation and survival, epithelial-to-mesenchymal transition, and invasion, angiogenesis, and metastasis [91,92,93]. Suppressed NF-κB signaling was detected in the colons of IQGAP2-deficient mice. The protein level of the p65 subunit of NF-κB was diminished compared to WT mice. Treatment with an inflammatory activator (dextran sulfate sodium), inducing p65 expression, had no effect in colons of IQGAP2-deficient mice [94]. However, transient knockdown of IQGAP2 had no effect on NF-κB baseline activation or p65 expression in liver cells, but significantly decreased the NF-κB promoter activation and p65 phosphorylation in response to type I interferons (IFN), indicating that the role of IQGAP2 in NF-κB activation is stimulus-dependent [90].

## 4. IQGAP2 and IQGAP3 Have Antithetic Roles in Many Types of Cancer

IQGAPs have been revealed to play critical roles in different malignancies, especially in the progression of cancer. Numerous researchers have observed that IQGAP2 and IQGAP3 play antagonistic roles in gastric cancer, colorectal cancer, hepatocellular carcinoma, prostate cancer, ovarian cancer, breast cancer, and malignant lymphoma. Lower expression levels of IQGAP2 or rather higher expression levels of IQGAP3 are associated with a worse prognosis (Table 2).

We have summarized the current evidence on the contribution of IQGAP2 and IQGAP3 to these different types of cancer as examples. Nevertheless, IQGAPs are also associated with other cancer types. These findings collectively support the concept that IQGAP2 and IQGAP3 act as critical regulators of tumorigenesis by scaffolding and promoting diverse oncogenic pathways.

### 4.1. Gastric Cancer

The relation between IQGAP2 and cancer was first mentioned in gastric cancer. Zhu et al. detected increased expression of IQGAP2, R-Ras, and B-Raf, and activation of the Erk1/2 pathway in virulence factor (CagA)-transformed immortalized gastric epithelial cells [71].

On the other hand, some reports indicate that IQGAP2 can act as a tumor suppressor in gastric cancer. IQGAP2 was found methylated and lost in gastric carcinoma tissues compared with gastric mucosa, and patients with IQGAP2 inactivation by methylation had a significantly worse prognosis. In an in vitro experiment, suppression of IQGAP2 increased cell invasion in gastric cancer cell lines, revealing its probable tumor suppressor role [32]. A subsequent study demonstrated that IQGAP2 may regulate gastric cancer through the AKT pathway [38], which is highly activated in nearly 80% of gastric cancers [105]. They found that IQGAP2 can bind to the PRD and SAM domains of SHIP2 in the cytoplasm of GC cells, resulting in an increase in SHIP2 phosphatase activity and thereby suppressing GC cell EMT via Akt inactivation.

The increase of IQGAP3 was observed in gastric cancer samples compared with normal controls, mostly distributed on the cell membrane and cytoplasm. Further examination of the clinicopathologic characteristics showed that IQGAP3 expression was significantly associated with the TNM classification and was an independent predictor of survival. IQGAP3 is also correlated with metastasis of gastric cancer, and overexpression enhances cell migration and invasion and reduces cell–cell adhesion [41,47].

### 4.2. Liver Cancer

The tumor suppressor role of IQGAP2 was first mentioned in hepatocellular carcinoma (HCC). There is substantial evidence that IQGAP2 plays a tumor suppressor role in HCC. Decreased IQGAP2 expression is related with increased tumor size, advanced tumor stage, worse tumor differentiation, as well as shorter postoperative tumor-free survival and overall survival after hepatectomy [95,96,97,106,107]. Further experiments in HCC cell lines demonstrated that IQGAP2 depletion may correlate with alpha-fetoprotein (AFP), the only current clinical biomarker for HCC. IQGAP2 was highly expressed in AFP+ cell lines, while it was lost in AFP– cell lines [107]. IQGAP2-deficient mice showed a high incidence of HCC compared with wildtype controls. Further analysis of these samples demonstrated a reduction of IQGAP1 expression in plasma membranes and an increase in IQGAP1 expression in the cytoplasm, and this pattern occurred in parallel with the activation of the Wnt/β-catenin pathway. The IQGAP1–IQGAP2-deficient mice were then generated to analyze the relationship between IQGAP1 and IQGAP2 in the progression of HCC. Survival durations of IQGAP1–IQGAP2-deficient mice were virtually equal to wildtype controls and were markedly enhanced compared to IQGAP2-deficient mice, which indicated that a loss of IQGAP2 may not be sufficient to induce a malignant phenotype; rather, this is dependent on the oncogenic IQGAP1 protein [35].

IQGAP3 was identified as an oncogene in HCC due to its overexpression in HCC tissues [86,98]. Its expression is associated with a larger tumor size, advanced tumor stage, and poor tumor differentiation. IQGAP3 induces intrahepatic and extrahepatic metastases in HCC, hence considerably lowering patient survival [86]. In addition, IQGAP3 is a new biomarker for HCC screening and diagnosis that detects small HCCs more effectively than AFP. IQGAP3 can be used as a biomarker in addition to AFP to increase the diagnostic accuracy of AFP-negative HCC. The combination of AFP, IQGAP3, and chaperonin containing TCP1 complex subunit 3 (CCT3) significantly increased the discriminatory ability of HCC compared to AFP alone [99].

### 4.3. Breast Cancer

Female breast cancer is the most commonly diagnosed cancer and the main cause of cancer-related mortality worldwide in women [108]. IQGAPs seem not to be correlated with different subtypes (ER, PR, HER2, or triple-negative) of breast cancer but associated with overall tumorigenic features.

Kumar et al. observed decreased IQGAP2 in most breast cancer tissue compared to normal tissue, and lower IQGAP2 expression was strongly correlated with lymph node metastases, lymphovascular invasion, and a higher cancer stage, but not with tumor size [40]. The expression pattern showed no clustering of specific molecular subtypes through the analysis on different breast cancer cell lines. Furthermore, reduced IQGAP2 expression leads to increased proliferation and reduced apoptosis regardless of ER status, which results in continuous tumor growth and development of breast cancer [40]. In search of the mechanism, the IQGAP2 promotes apoptosis via activating the p38-p53 pathway, triggered by an increase in reactive oxygen species (ROS). Invasion of breast cancer is mainly a cumulative result of EMT and angiogenesis [109,110]. Kumar et al. recently showed that depletion of IQGAP2 promoted both EMT via activation of the ERK pathway and angiogenesis in a paracrine manner through activating VEGFR2-AKT [40,82]. Similarly, Wang et al. found that IQGAP2 can suppress the proliferation, migration, and invasion of TNBC cells [39], but they did not observe any influence on the apoptosis. How IQGAP2 expression is regulated is mostly unknown. In breast cancer, the miR-10b-5p, which has been reported to be dysregulated and linked to prognosis, could negatively target IQGAP2 [39].

In comparison to IQGAP2, IQGAP3 is highly expressed in breast cancer tissues relative to adjacent normal tissues [49,73]. Overexpression of IQGAP3 was substantially linked with a higher clinical stage, distant metastases, locoregional recurrence, and radio-resistance. In addition, IQGAP3 is an independent negative prognostic factor for breast cancer. Patients with elevated levels of IQGAP3 had a poor prognosis, even after radiation [49].

### 4.4. Prostate Cancer

The expression of IQGAP2 in prostate cancer is contradictory. Analysis of the transcriptome revealed that IQGAP2 was overexpressed in prostate cancer tissues compared with normal adjacent prostate tissue [100,111]. Meanwhile, IQGAP2 expression was found elevated in low-grade tumors (prostatic intraepithelial neoplasia to Gleason 3 tumors), which could be applicable for PC diagnosis in the future, as its accuracy for detection was higher than the PSA level. On the other hand, IQGAP2 was downregulated in high-grade tumors (Gleason 4–5) [33,37]. Survival analysis of the public database revealed that downregulation of IQGAP2 in prostate cancer is positively correlated with recurrence and metastasis [37]. Further molecular experiments confirmed the suppressor role of IQGAP2 in prostate cancer. IQGAP2 inhibited cell proliferation and invasion of prostate cancer cell lines through enhancing E-cadherin promoter activity via inhibiting AKT activation [33].

Analyses of TCGA-PRAD (prostate adenocarcinoma) data revealed 733 overexpressed genes, including IQGAP3, which is also frequently mutated [112]. The higher expression of IQGAP3 was indicative of a worse overall survival and disease-free interval (DFI) [113]. Moreover, the expression of IQGAP3 was positively correlated with infiltration of B cells, macrophages, and dendritic cells, indicating its potential role as a tumor-specific antigen and therapeutic target in PRAD [113].

### 4.5. Bladder Cancer

Bladder cancer is the tenth most commonly diagnosed cancer in the world. The gold standard diagnostic method for bladder cancer is the cystoscopy, an invasive, uncomfortable, and costly procedure, which is also used for postoperative follow-up [114]. Sufficiently sensitive urine-based markers are not available. In recent years, the urinary nucleic acids have been widely examined, which are directly derived from the urinary tract tumor cells’ death [115]. Won et al. detected overexpression of IQGAP3 in urine samples from bladder cancer patients compared with normal controls. ROC curve analysis revealed that IQGAP3 is a highly sensitive and specific diagnostic marker for bladder cancer, which can reach 80.0% and 83.8%, respectively. The specificity in hematuria was up to 90.7% [116,117]. IQGAP3 was also overexpressed in bladder cancer tissues, and can promote bladder cancer progression via activating the Ras/ERK pathway [43].

Although almost all studies have confirmed that IQGAP1 is an oncogene, IQGAP1 was reported as a tumor suppressor in bladder cancer [118], playing a completely opposite role compared with other cancers. IQGAP1 depletion increases tumor growth in vitro and in vivo via TGF-β signaling and is associated with a worse prognosis in bladder cancer patients.

In a recent study, we conducted research on IQGAP2 in bladder cancer for the first time [119]. Bioinformatic analysis using TCGA and published data showed reduced IQGAP2 in bladder cancer tissue compared with normal tissue. We also observed decreased IQGAP2 mRNA and protein in bladder cancer cell lines compared with normal urothelium cell lines [119]. Then, we analyzed the functional role of IQGAP2 in bladder cancer cells. The knockdown of IQGAP2 promoted proliferation, migration, and invasion, while the overexpression of IQGAP2 had the opposite role. Pathway mapping using TCGA data and confirmatory experiments indicated that this effect was achieved through the regulation of the MAPK/ERK pathway and cytokines [119].

### 4.6. Ovarian Cancer

As in gastric cancer, hypermethylation-mediated inactivation of IQGAP2 was discovered in serous and clear cell ovarian cancer samples, and it was associated with worse progression-free survival. The pull-down assay in ovarian cells confirmed that IQGAP2 did not affect the activity of Ras, but altered the phosphorylation of AKT, ERK, and the protein expression, as well as its nuclear translocation of beta-catenin. Further analysis proved that IQGAP2 inhibited the migration, invasion, and EMT of ovarian cancer cells through deactivation of Wnt/-catenin signaling [34].

IQGAP3 expression was upregulated in high-grade serous ovarian cancer samples compared with the fallopian tubal samples, and this increase was significantly associated with poor overall survival, as well as progression-free survival [45]. The knockdown of IQGAP3 inhibited the metastasis of high-grade serous ovarian cancer (HGSOC) in vitro and in vivo. IQGAP3 regulated cell proliferation via regulation of apoptosis in ovarian cancer cells [45]. IQGAP3 was also found to correlate with medication sensitivity of Olaparib, a PARP inhibitor. The knockdown of IQGAP3 increases the sensitivity of ovarian cancer cells to Olaparib, which may be accomplished by the regulation of proteins associated with DNA damage and chemoresistance. Furthermore, the knockdown of IQGAP3 also influenced a series of oncogenic mechanisms in ovarian cancer, as demonstrated by the alteration in the protein expression, including EMT-related proteins (E-CAD, N-CAD, ZEB-1, Vimentin, and Snail) and apoptosis-related proteins (Caspase-3, Caspase-9, Bcl2, and Bax) [45].

### 4.7. Colorectal Cancer

The mechanism of IQGAP2 was not analyzed in detail in colorectal cancer (CRC). Dinesh Kumar reported that IQGAP2 was decreased in CRC tissue, both at the mRNA and protein levels. They did not see any significant correlation with OS [36]. Anyway, another report found that IQGAP2 was overexpressed in tissues of CRC [102]. Two publications revealed how IQGAP2 was regulated. The overexpression of miR-92a and miR-29a-3p can negatively regulate IQGAP2 in CRC cell lines [120,121]. However, considering that dysregulation of Wnt signaling is necessary for the development of colorectal cancer, and nearly all CRCs exhibit abnormalities in Wnt signaling, it will be interesting to clarify if IQGAP2 plays a role in the carcinogenesis of colorectal cancer through the Wnt pathway in further research [34,35,79,122].

IQGAP3 was elevated at the tissue and cellular levels in colorectal cancer [42,48,103]. Increased IQGAP3 expression was associated with a higher tumor stage and shorter overall survival. Moreover, Cao et al. also detected higher IQGAP3 levels in the serum of CRC patients than those in the healthy group, with higher sensitivity than carcinoembryonic antigen (CEA) and cancer antigen 19-9 (CA19-9), which are the most widely used diagnostic markers in CRC [101,103,123].

### 4.8. Malignant Lymphoma

Beyond solid tumors, IQGAPs also play roles in malignant lymphoma. Interestingly, they played the exact opposite roles to that in solid tumors. Diffuse large B cell lymphoma (DLBCL) accounts for 25–35% of non-Hodgkin lymphomas and is the most prevalent kind of lymphoma worldwide [124,125]. Although 50–70% of patients are usually treated with cyclophosphamide, doxorubicin, vincristine, and prednisone (CHOP) chemotherapy and combined immunochemotherapy with rituximab (R-CHOP), approximately one-third of individuals with DLBCL suffer from highly deadly recurring or progressive illnesses, so identifying these patients early, and providing alternative therapy, is critical [126,127].

IQGAP2 mRNA expression is significantly higher in hematologic malignancies than in solid tumors. Furthermore, high-grade lymphoma expressed higher IQGAP2 protein levels than low-grade lymphoma, indicating that IQGAP2 is related to the malignancy of DLBCL. On the other hand, DLBCL patients with elevated IQGAP2 mRNA had reduced survival periods, even when treated with CHOP and R-CHOP chemotherapy. IQGAP2-associated gene enrichment analysis revealed a high link between IQGAP2 and immunological processes. IQGAP2 mRNA expression was positively linked with immunosuppressive genes and infiltration of leukocytes, suggesting that IQGAP2 may be implicated in immunosuppression in DLBCL, which is a crucial factor in DLBCL carcinogenesis [104].

Despite that IQGAP3 was found as the most conspicuously overexpressed gene in DLBCL samples relative to normal controls, patients with high IQGAP3 expression had a much better clinical outcome, as indicated by longer progression-free (PFS) and overall survival (OS) [75]. In patients whose lymphoma cells lacked active PI3K signaling, the expression of IQGAP3 did not predict any survival outcome. However, in patients with an activated PI3K pathway, a higher IQGAP3 level predicted a significantly better clinical outcome. In addition, IQGAP3 could inhibit RAS activity in DLBCL cells, hence dramatically limiting DLBCL cell motility. The data reveal that IQGAP3 is one of the most essential molecules that connect the RAS and PI3K pathways, allowing them to influence each other and co-regulate functions farther down, and that this cross-talk contributes to the development and progression of cancer [75].

## 5. Putative Explanation for IQGAPs’ Opposite Functions

The paradoxical fact that IQGAPs play opposing roles in malignancies may be attributable to the slight differences in the similar domain structure and sequence homology. For instance, the calcium-sensing protein CaM (calmodulin) attaches to all the IQGAPs via IQ motifs, but its stability varies between each member. In the presence of calcium ions, IQGAP2 binds to CaM with a lower affinity than IQGAP1 and 3. This difference has been explained in that CaM interacts with IQGAP2 via the second and third binding sites of the IQ motif, whereas IQGAP1 and 3 connect with CaM via all four IQ motifs. However, the binding between IQGAP2 and CaM is temporary, whereas IQGAP3 displays long-lived binding with its last three IQ motifs and transient binding with its first IQ motif [62].

Similarity, the Rho family of small GTPases showed different binding affinity to IQGAPs. IQGAP1 and 3 have higher binding affinity for the active GTP-Cdc42 and GTP-Rac1 than for the inactive, GDP-bound form of the GTPases, through the GRD domain [50]. However, the IQGAP2 has been observed to interact without discrimination with both the GDP and GTP-bound versions [31,128]. In addition, IQGAP3 strongly interacted with the active form of Ras, but only moderately with the wildtype or dominant-negative form. IQGAP1 and 2 could not stimulate the GTPase activity of Ras [31].

The binding affinity of IQGAPs to the essential light chain of myosin and S100B also differs. IQGAP1 has a high affinity for binding to myosin essential light chain and S100B, but IQGAP2 and IQGAP3 have transitory or no affinity for binding to these proteins [62,129,130]. In addition, Anillin proteins co-immunoprecipitate with IQGAP3 but not with IQGAP1 or IQGAP2, which suggests the importance in cytokinesis [131]. Therefore, further in-depth studies on the molecular binding and structures of IQGAPs should be completed to uncover unresolved issues, such as why IQGAPs exert opposing effects on the MAPK/ERK and PI3K-AKT signaling pathways.

## 6. Conclusions

In the past decade, significant evidence on the function of IQGAPs in signal transduction during carcinogenesis has emerged. This review described the architecture of IQGAP2 and IQGAP3, as well as their expression and antithetic roles through relevant signaling pathways in various malignancies.

IQGAP2 expression is reduced and plays a tumor suppressor role in most solid cancer types, while IQGAP3 is overexpressed and acts as an oncogene in the same cancer types. Only in lymphoma did DLBCL patients with a higher expression of IQGAP2 have a worse prognosis, and IQGAP3 overexpression was correlated with an excellent prognosis.

It has been proven that the alteration of IQGAPs’ expression in serum or urine can differentiate HCC, CRC, and bladder cancer from healthy patients. Thus, IQGAPs may be applicable as promising biomarkers. Different studies indicated that the reduction of IQGAP2 may be correlated with the abnormal promoter methylation. Further studies on the epigenetic state of IQGAP2 may obtain some promising results. In addition to methylation, the other factors that trigger the altered expression of IQGAPs in cancer also need further investigation. There is now considerable evidence that IQGAPs regulate a multitude of pathways to modulate cancer processes and chemoresistance, but some questions, such as how they trigger this signaling, through which domains, and why they play opposite roles on the same pathways, are still unanswered. Thus, it is crucial to investigate the dynamics between IQGAP-mediated pathways in cancer. With a greater understanding of the role of IQGAPs in cancer and the underlying mechanisms, drugs targeting IQGAPs and their related signaling could hold promise.

## Figures and Tables

**Figure 2 cancers-15-01115-f002:**
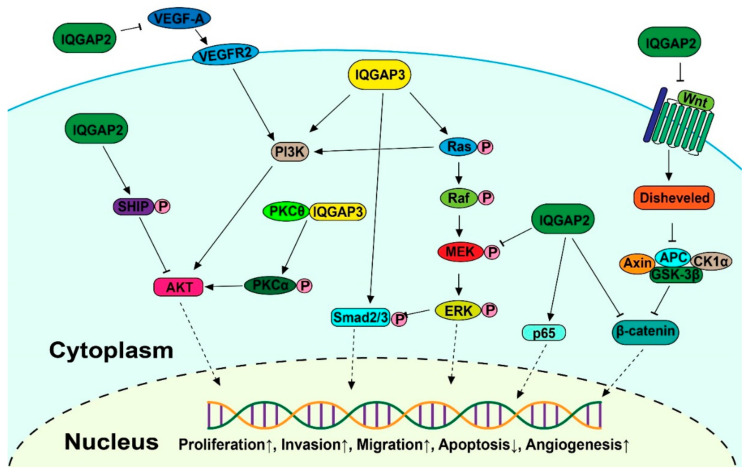
Schematic diagram of the IQGAP-mediated signaling pathways involved in carcinogenesis (Permission obtain from OA journal: [67]).

**Table 1 cancers-15-01115-t001:** The protein expression of IQGAPs in organs (data adapted from the Human Protein Atlas database: https://www.proteinatlas.org/, accessed on 15 October 2022) [53].

	IQGAP1	IQGAP2	IQGAP3
Breast	+++	+	++
Brain	+	++	+++
Bladder	++	-	+++
Colon	+++	++	+++
Kidney	+++	++	+++
Liver	++	+	++
Lung	+++	++	+
Prostate	++	+++	++
Stomach	++	+++	+++
Thyroid gland	++	++	+++
Testis	+	+	+++
Skin	+++	+	+++

**Table 2 cancers-15-01115-t002:** Cancers with altered IQGAP2 and IQGAP3 expression. IHC: immunohistochemistry; LC-MS: liquid chromatography–mass spectrometry; WB: Western blot; RT-PCR: reverse transcription-polymerase chain reaction; ELISA: enzyme-linked immunosorbent assay; PIN: prostatic intraepithelial neoplasia; DLBLC: diffuse large B cell lymphoma; GCB: germinal center B cell; ABC: activated B cell.

Cancer Type	Comparison	Expression Alteration	Sample	Method	Prognostic Relevance	Reference
Liver	Carcinoma vs. normal Carcinoma vs. normal Carcinoma vs. normal	IQGAP2↓	Tissue	IHC LC-MS/WB/IHC RT-PCR/WB/IHC	/ / Yes	[95] [96] [97]
	Carcinoma vs. adenoma	IQGAP3↑	Tissue Serum	RT-PCR/WB/IHC DNA sequencing IHC ELISA	Yes / Yes Yes	[86] [98] [83] [99]
Prostate	Carcinoma vs. normal PIN/Gleason ≤ 3 vs. normal Gleason 4–5 vs. PIN/Gleason ≤ 3	IQGAP2↑ IQGAP2↑ IQGAP2↓	Tissue	RT-PCR IHC IHC	/	[100] [33] [33]
	Carcinoma vs. normal	IQGAP3↑	Tissue	DNA sequencing	Yes	[101]
Breast	Carcinoma vs. normal	IQGAP2↓	Tissue	IHC RT-PCR/WB	Yes /	[40] [39]
	Carcinoma vs. normal	IQGAP3↑	Tissue	RT-PCR/WB RT-PCR/WB/IHC	/ Yes	[73] [49]
Gastric	Carcinoma vs. normal	IQGAP2↓	Tissue	IHC	Yes	[32]
	Carcinoma vs. normal	IQGAP3↑	Tissue	IHC IHC	Yes No	[47] [41]
Ovary	Serous and clear cell ovarian cancer vs. normal	IQGAP2↓	Tissue	DNA sequencing /WB	Yes	[34]
	High-grade serous ovarian cancer vs. normal	IQGAP3↑	Tissue	RT-PCR/WB/IHC	Yes	[45]
Colorectal	Carcinoma vs. normal	IQGAP2↓ IQGAP2↑	Tissue	IHC DNA sequencing	No /	[36] [102]
	Carcinoma vs. normal	IQGAP3↑	Tissue/Serum Tissue	IHC/ELISA RT-PCR/IHC RT-PCR/IHC	Yes Yes Yes	[103] [48] [42]
Bladder	Carcinoma vs. normal	IQGAP2 (Heterogeneity)	Tissue	IHC		[43]
	Carcinoma vs. normal	IQGAP3↑	Tissue Tissue/Urine	RT-PCR/IHC RT-PCR	/ /	[43] [46]
DLBLC	GCB DLBCL vs. ABC DLBCL	IQGAP2↓	Tissue	DNA sequencing	Yes	[104]
	Cancer vs. normal	IQGAP3↑	Tissue	DNA sequencing	Yes	[75]

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
