# Peer review of "The Antithetic Roles of IQGAP2 and IQGAP3 in Cancers"

_cancers, 2023, doi:10.3390/cancers15041115_

Round 1
Reviewer 1 Report
The most recent review of” IQGAP2 and IQGAP3” was “the biology of IQGAP proteins: beyond the cytoskeleton”, which was published in 2015 (EMBO Rep 2015 Apr;16(4)). In 2021, Wei T et al reviewed the “Role of IQGAP1 in Carcinogenesis”, which was published in Cancers (Basel), 2021 Aug 04;13(16). Since the role of IQGAP2 and IQGAP3 in cancer were analyzed less, the subject of this article is noteworthy. Taking the article (Cancers (Basel) 2021 Aug 04;13(16)) as reference, the author elaborated on the antithetic roles of IQGAP2 and IQGAP3 that play in multiple key pathways and various forms of cancers, respectively. The organizational structure is acceptable. However, this paper has the following issues.
1. The aim of this review was to summarizes the antithetic roles of IQGAP2 and IQGAP3 in cancers,but only more than 40 of total 141 references cited in this review were directly related to the IQGAP2 and IQGAP3. It seems that there are not enough articles, up to now, to support this review. More direct evidence on the IQGAP2 and IQGAP3 needs to be supplied. Furthermore, lots of content beyond the subject dilute the theme of the article, for example the role of the IQGAP1. Even if part of them is relevant or necessary to the paper, the material needs to be restated.
2. This article summaries the literature from two aspects, in different pathways and various types of tumors, but some formulation is confused and not well differentiated in these two parts. Moreover, the review should organize the reference well to serve the author's point of view on the basis of listed materials. Inappropriate article organization make some conclusions seem blunt, such as “~~~utilizing IQGAPs as treatment targets for cancers” or “~~~as biomarkers”.
3. It is inappropriate to use unpublished research results as the reference, such as in the part of bladder cancer.
Taken together, this review needs to be reorganized around the theme to make it more concise and clearer, which will assist the author to draw corresponding conclusions.
Author Response
Dear Reviewer #1,
Thank you for reviewing our manuscript, your comments and constructive criticism. We hope that we have been able to answer all of your comments satisfactorily.
- The aim of this review was to summarizes the antithetic roles of IQGAP2 and IQGAP3 in cancers,but only more than 40 of total 141 references cited in this review were directly related to the IQGAP2 and IQGAP3. It seems that there are not enough articles, up to now, to support this review. More direct evidence on the IQGAP2 and IQGAP3 needs to be supplied. Furthermore, lots of content beyond the subject dilute the theme of the article, for example the role of the IQGAP1. Even if part of them is relevant or necessary to the paper, the material needs to be restated.
You are right, there are only few articles covering IQGAP2 and IQGAP3. However, the studies published in the last 10 years have been very promising, but more direct evidence on IQGAP2 and IQGAP3 is not available. So, we are taking the plunge ourselves, and by publishing this review we are calling on more researchers to join in.
Because much off-topic content diluted the theme, we revised the manuscript extensively and omitted some statements, not related directly to IQGAPs. This reduced the number of citations to 127.
- This article summaries the literature from two aspects, in different pathways and various types of tumors, but some formulation is confused and not well differentiated in these two parts. Moreover, the review should organize the reference well to serve the author's point of view on the basis of listed materials. Inappropriate article organization make some conclusions seem blunt, such as “~~~utilizing IQGAPs as treatment targets for cancers” or “~~~as biomarkers”.
Thank you for this hint. The signaling pathways that are affected by IQGAPs are mainly associated with the effect of IQGAPs in different cancer types. Therefore, it is difficult to accurately separate these two parts. Nevertheless, we have tried to avoid detailing the signaling pathway in the sections on the role in different cancer entities in the revised manuscript.
Inappropriate conclusions about the use of IQGAPs as a biomarker or therapy target are rephrased or deleted in the revised manuscript due to lack of evidence.
- It is inappropriate to use unpublished research results as the reference, such as in the part of bladder cancer.
Thank you for your reminder. We have realized that this is not scientific and have deleted the unpublished results in bladder cancer part.
Reviewer 2 Report
The review by Song et al describes the role of IQGAP2 and IQGAP3 in various cancers. While the role of IQGAP1 in cellular signaling has been extensively studied, less is known about IQGAP2 and IQGAP3. Here the authors provide insights into how IQGAP 2 and 3 can have opposing effect on cancer growth and EMT. It is an interesting article but is written poorly, needs thorough editing for grammatical errors, repetitions, and contradictory statements. A few necessary edits are shown in the specific comments below:
Specific Comments:
1. The apostrophes are used wrongly in many places, including in abstract, when mentioning IQGAP's role.
2. Repetition: Line 21 and 70; line 56 and 65
3. Statements such as IQGAP was ‘positive correlated’’ need to be changed to ‘positively correlated’.
4. Usage of the abbreviation ‘BC’ for bladder cancer can be confused for breast cancer.
5. Line 310: Sentence starting ‘Furthermore, …….increased breast cancer growth regardless of ER status….., which is associated with apoptosis via deactivating p38-p53 pathway,….’. To this reviewer this statement suggests that breast cancer growth is associated with apoptosis. The authors might want to reword this sentence.
6. Example of contradictory statement: In line 176 the authors state that "It has been discovered that IQGAP2 increases E-cadherin expression and ‘promotes’ EMT via decreasing Akt activation in prostate cancer”. In line 341 they state that “IQGAP2 ‘inhibited’ cell proliferation and invasion of prostate cancer cell line through enhancing E-Cadherin promoter activity via inhibiting Akt activation”. These kind of discrepancies need to be identified and rectified throughout the article.
These are just a few suggestions, there are several such mistakes all through the article that needs to be taken care of.
Author Response
Dear Reviewer 2,
Thank you for reviewing our manuscript, your comments and constructive criticism. To address all criticism and comments, we extensively revised our manuscript. We hope that we have been able to answer all of your comments satisfactorily.
- The apostrophes are used wrongly in many places, including in abstract, when mentioning IQGAP's role.
Thanks for your carefulness, we corrected this mistake.
- Repetition: Line 21 and 70; line 56 and 65
We carefully checked the manuscript and deleted all repetition in our revised manuscript.
- Statements such as IQGAP was ‘positive correlated’’ need to be changed to ‘positively correlated’.
We revised this careless grammatical error.
- Usage of the abbreviation ‘BC’ for bladder cancer can be confused for breast cancer.
Thanks for your suggestion. We replaced ‘BC’ with ‘bladder cancer’.
- Line 310: Sentence starting ‘Furthermore, …….increased breast cancer growth regardless of ER status….., which is associated with apoptosis via deactivating p38-p53 pathway,….’. To this reviewer this statement suggests that breast cancer growth is associated with apoptosis. The authors might want to reword this sentence.
Thanks for hint. According to previous reports, the apoptosis is associated with tumor growth. We reword the sentence to:
Furthermore, reduced IQGAP2 expression leads to increased proliferation and reduced apoptosis regardless of ER status, which result in continuous tumor growth and development of breast cancer [40]. In search of the mechanism, the IQGAP2 promotes apoptosis via activating p38-p53 pathway, triggered by increase in reactive oxygen species (ROS).
- Example of contradictory statement: In line 176 the authors state that "It has been discovered that IQGAP2 increases E-cadherin expression and ‘promotes’ EMT via decreasing Akt activation in prostate cancer”. In line 341 they state that “IQGAP2 ‘inhibited’ cell proliferation and invasion of prostate cancer cell line through enhancing E-Cadherin promoter activity via inhibiting Akt activation”.
We made a mistake in line 176, it should be ‘It has been discovered that IQGAP2 increases E-cadherin expression and inhibits EMT via reduction of Akt activation in prostate cancer [33]’.
- These kind of discrepancies need to be identified and rectified throughout the article. These are just a few suggestions, there are several such mistakes all through the article that needs to be taken care of.
We sincerely apologize for the above mistakes. We thoroughly reviewed the manuscript and corrected these issues.
Round 2
Reviewer 1 Report
The authors satisfactorily answered the questions of this reviewer.
Reviewer 2 Report
The authors have addressed the concerns raised by this reviewer.